

# Metagenomic 16S rDNA reads of *in situ* preserved samples revealed microbial communities in the Yongle blue hole

Hongxi Zhang[1,2], Taoshu Wei[1], Qingmei Li[1], Liang Fu[3], Lisheng He[1] and Yong Wang[4]

[1] Institute of Deep Sea Science and Engineering, Chinese Academy of Sciences, Sanya, Hainan, China
[2] University of Chinese Academy of Sciences, Beijing, China
[3] Sansha Trackline Institute of Coral Reef Environment Protection, Sansha, Hainan, China
[4] Institute for Ocean Engineering, Shenzhen International Graduate School, Tsinghua University, Shenzhen, China

Corresponding author
Yong Wang,
wangyong@sz.tsinghua.edu.cn

## ABSTRACT

Our knowledge on biogeochemistry and microbial ecology of marine blue holes is limited due to challenges in collecting multilayered water column and oxycline zones. In this study, we collected samples from 16 water layers in Yongle blue hole (YBH) located in the South China Sea using the *in situ* microbial filtration and fixation (ISMIFF) apparatus. The microbial communities based on 16S rRNA metagenomic reads for the ISMIFF samples showed high microbial diversity and consistency among samples with similar dissolved oxygen levels. At the same depth of the anoxic layer, the ISMIFF samples were dominated by sulfate-reducing bacteria from Desulfatiglandales (17.96%). The sulfide concentration is the most significant factor that drives the division of microbial communities in YBH, which might support the prevalence of sulfate-reducing microorganisms in the anoxic layers. Our results are different from the microbial community structures of a Niskin sample of this study and the reported samples collected in 2017, in which a high relative abundance of Alteromonadales (26.59%) and Thiomicrospirales (38.13%), and Arcobacteraceae (11.74%) was identified. We therefore demonstrate a new profile of microbial communities in YBH probably due to the effect of sampling and molecular biological methods, which provides new possibilities for further understanding of the material circulation mechanism of blue holes and expanding anoxic marine water zones under global warming.

## INTRODUCTION

Marine blue holes are deep caves formed by sinking carbonate rock bodies in the ocean (*Martin, Gulley & Spellman, 2012*; *Mylroie, Carew & Moore, 1995*; *van Hengstum et al., 2019, 2020*), with their entrance close to the sea surface. The water column of some marine blue holes includes an oxic layer, a relatively stable oxic-anoxic interface, and an anoxic layer, while others do not have an anoxic water body (*Iwanowicz et al., 2021*; *Kindler et al., 2022*; *Patin et al., 2021*). Organisms living under different dissolved oxygen concentrations

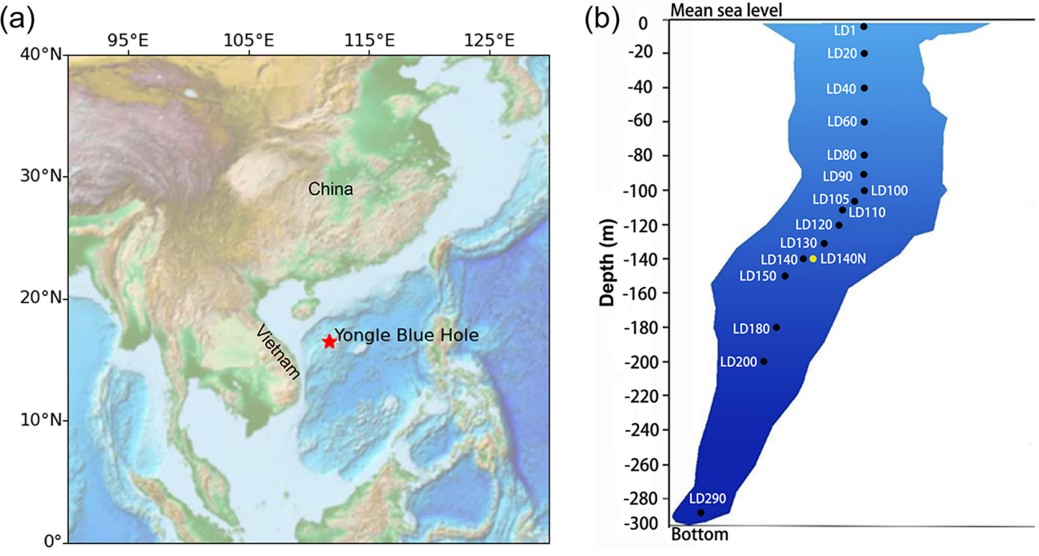

**Figure 1 Sampling in the Yongle blue hole.** The location of the Yongle blue hole in the South China Sea (A) and the vertical distribution of the sampling sites (B). Sample IDs refer to Table S1. The ISMIFF samples are in black, and the Niskin sample is in yellow. Map was created using the ETOPO (Earth TOPOgraphy) data. Data source: national centers for environmental information (NCEI), NOAA. This map is in the public domain and is not subject to copyright restrictions.

in the blue holes are precious samples for studying adaptation and metabolic patterns in relatively isolated, oxycline environments. The Amberjack blue hole in the Gulf of Mexico and the Yongle blue hole (YBH) in the Xisha (Paracel) Islands (Fig. 1A) both have oxic and anoxic layers, as well as a large number of novel microbes (*Patin et al., 2021*; *Zhang et al., 2021*). YBH is currently known as the deepest blue hole on Earth with a depth of 300.89 m (*Yao et al., 2020*). The oxic water of YBH is above ~100 m with a seasonal changing depth, while below ~100 m is an anoxic zone that contains a large amount of hydrogen sulfide and methane possibly produced by biodegradation (*Xie et al., 2019*). The input of oxygen from the outside seawater to YBH is limited due to weak vertical water cycling, which resulted in a rapid decrease of the oxygen content to about 6 mg/L at 10-m depth. At about 200 m, the axis of YBH is offset by about 60 m (Fig. 1B), so that the outside seawater and particulate organic matter cannot directly reach the bottom (*Gao et al., 2022*; *Li et al., 2020, 2018*). This topographic feature further increases the environmental stability of the YBH bottom and also limits the influence of the external environment on the creatures in it. According to the physicochemical parameters of YBH, the water environment below 160 m is basically in a stable state (*Peng et al., 2018*). In comparison, the Amberjack Hole in the Gulf of Mexico contains an anoxic layer with only about 40-m depth, and external organic carbon can precipitate directly to its bottom (*Patin et al., 2021*). Although the anoxic zone of the Black Sea (*Cabello-Yeves et al., 2021*), Cariaco Basin (*Mara et al., 2020*), and Eastern Tropical South Pacific (*Karstensen, Stramma & Visbeck, 2008*) are large, they are not conducive to the formation of semi-closed ecosystems due to their wide openness to

external water bodies. Thus, YBH has a semi-enclosed environment and a long-term stable structure, which could probably have driven the formation of a unique ecosystem in YBH.

Microbiomes may be stratified by environmental gradients (*DeLong et al., 2006*). Using amplicons of 16S and 18S ribosomal genes, the microbial community structures in the water column at different depths of YBH were preliminarily analyzed (*He et al., 2019*; *Liu et al., 2019*). Among them, the archaea are mainly affiliated with Thaumarchaeota and Euryarchaeota, comprising a total of three phyla and at least 1,010 species, which is higher than that found in the seawater outside the hole. The bacteria are dominated by Chloroflexi and Proteobacteria with a total of about 53 phyla and 301 genera. Microbial processes such as nitrate reduction, sulfur oxidation, sulfur reduction, methanogenesis, and ammonia oxidation were detected in the anoxic zone of YBH by amplification of functional genes of the microorganisms involved in sulfur and nitrogen cycles. A large number of sulfur oxidation and nitrification genes were detected in the suboxic and anoxic zones of the blue hole. The distribution of these genes and microorganisms was explained to some extent by vertical environmental changes in the water columns of the blue holes. However, the inconsistency of microbial communities in anoxic layers remains a question (*He et al., 2019*; *Liu et al., 2019*). As an example, the Amberjack blue hole has a relatively stable microbial population in a redox-stratified environment (*He et al., 2020*; *Patin et al., 2021*). In previous reports, *Alteromonas* was the main genus in the oxic zone of YBH, likely degrading organic matter and performing nitrate reduction; SUP05 bacteria rely on oxygen and nitrate to oxidize hydrogen sulfide, preventing hydrogen sulfide from being spilled from YBH (*He et al., 2020*). In the YBH oxic-anoxic transition zone, anaerobic ammonia-oxidizing (anammox) bacteria have been detected. These microorganisms supported a unique ecosystem in YBH under semi-enclosed conditions, which might rely on organic matter input from the top layer and assimilation of inorganic carbon derived from organic carbon mineralization to synthesize organic matter (*Cifuentes et al., 2021*; *He et al., 2019*, *2020*; *Xie et al., 2019*; *Yao et al., 2020*; *Zhang et al., 2021*). At present, the biodiversity and taxonomic profile of the anaerobic microorganisms in YBH have not been fully understood.

In this study, we obtained *in situ* filtered and well-preserved water samples from YBH using ISMIFF. This sampling methods might minimize effects of onboard sample treatments on the YBH microbes. By analyzing the 16S rRNA metagenomic reads (miTags), the community structures of water samples collected by ISMIFF and Niskin bottles were compared. The results showed prokaryotic microbial community structures across the water column of YBH, which are distinct to the previous studies with remarkable differences in relative abundance of the major microbial inhabitants. The distinct microbial community profile allows for new understanding of the ecological functions of the microbes dwelling in the anoxic water of YBH.

## MATERIALS AND METHODS

### *In situ* sampling

Water samples were collected from YBH (16°31′30″N; 111°46′05″E) in April 2021 (Fig. 1). ISMIFF apparatus (*Wang et al., 2019*) was used to collect 16 *in-situ* filtered microbial

samples from different depths of YBH (Table S1). The filtration volume of seawater from each depth was ~30–60 L. Before filtration on a 0.22-μm pore size polycarbonate membrane (Millipore, Burlington, MA, USA), the water was prefiltered through a 30-μm pore size membrane. A Niskin bottle (Sea-Bird, Bellevue, WA, USA) was attached to an SBE 37SMP-ODO conductivity-temperature-depth (CTD) unit (Sea-Bird, Bellevue, WA, USA) to obtain a control water sample at 140 m depth. During this operation, a small remotely operated vehicle (ROV) carrying the Niskin bottle moved slowly to avoid the disturbance to particles in water and sediment. Immediately after the ISMIFF was recovered, the membrane was taken out from the filtration chamber and put into a 15-ml centrifuge tube. After the addition of 3 ml RNAlater (Ambion, Carlsbad, CA, USA), the tube was stored at −80 °C until further processing. The water sample collected by the Niskin bottle was filtered onboard through a 0.22-μm pore size polycarbonate membrane (Millipore, Burlington, MA, USA). The membranes were added 3 ml RNAlater and then maintained at −80 °C as well.

Profiles of temperature and dissolved oxygen (DO) were determined using the SBE 37SMP-ODO conductivity-temperature-depth (CTD) unit and a DO sensor (Sea-Bird, Bellevue, WA, USA). pH and salinity were measured directly with the Hach model HQ 40d portable combination meter, and $H_2S$ was measured with a Hach DR900 colorimeter (HACH, Loveland, CO, USA).

## Nucleic acids extraction and metagenome sequencing

Total genomic DNA was extracted from 16 ISMIFF and one Niskin sample using DNA/RNA co-extraction kit (Tiangen, Beijing, China) following the manufacturer's instructions. DNA concentrations were measured using a Qubit™ 2.0 Fluorometer (Invitrogen, Carlsbad, CA, USA). A total of 100 ng DNA was used as a template for the construction of a metagenomic high-throughput sequencing library. In the process of library construction, the long DNA fragments were ultrasonically broken by the Covaris M220 instrument (Covaris, Woburn, Massachusetts, USA), and the length of the inserted fragments in the library was about 350 bp. Then, the metagenomic high-throughput library was constructed by using VAHTS Universal DNA Library Prep Kit for Illumina V3 (Vazyme, Nanjing, China) according to the instructions. The metagenomic libraries were sequenced using an Illumina Novaseq 6000 platform (2 × 150 bp).

## Bioinformatic analysis

The reference metagenomic raw data for the previous Yongle blue hole microbial studies (*He et al., 2020*) were downloaded from the NCBI SRA database (Table S2). FastQC (v0.11.8) (http://www.bioinformatics.babraham.ac.uk/projects/fastqc/) was used to evaluate the quality of the raw Illumina sequencing data of the 17 metagenomes. Raw reads were trimmed to remove adapters and then filtered using fastp (v0.23.1) (*Chen et al., 2018*) with parameters (−w 24 − c − q 20 − u 20 − g − W 5 − 3 − l 50). Low quality reads (assigned by a quality score <20 for >20% of the read length), those shorter than 50 bp, and unpaired were removed. Data with a high level of duplications (>5%) were pre-processed with FastUniq (v1.1) (*Xu et al., 2012*) to remove duplicated reads.

Prokaryotic ribosomal RNA miTags (5S, 16S, and 23S) were identified from the clean metagenomic reads using rna_hmm3.py (*Huang, Gilna & Li, 2009*), which employed HMMER (v.3.1b2) (*Mistry et al., 2013*) to predict ribosomal RNA gene fragments from both forward and reverse metagenomic reads. 16S miTags (≥100 bp) were recruited by an in-house python script (*Zhou et al., 2022*) and were matched to the variable region V4 of the 16S rRNA gene by HMMsearch software (*Eddy, 2011*) against a V4 Hidden Markov Model. The 16S miTags were imported into QIIME2 (v.2022.2) (*Bolyen et al., 2019*) with the setting of—type 'SampleData [Sequences]' and clustered into operational taxonomic units (OTUs) with ≥97% similarity using the VSEARCH pipeline (*Rognes et al., 2016*). The tag sequence with the highest abundance was selected as the representative sequence for each cluster. Feature-classifier classify-sklearn command in QIIME2 was used to classify the OTUs in reference to SILVA SSU 138.1 database (*Glockner et al., 2017*; *Quast et al., 2013*). Moreover, the taxa at order level were used for non-metric Multidimensional scaling (NMDS) and canonical correlation analysis (CCA). ANOSIM (analysis of similarities) based on the Bray-Curtis distance was used to calculate the pairwise similarities of the microbial community compositions at order level.

## Phylogenetic analysis

We obtained reference 16S rRNA gene sequences for Desulfatiglandales, Alteromonadales, Thiomicrospirales and Campylobacterales from the NCBI database. The sequencing read representing a target 16S miTag OTU was selected and then pooled into a dataset containing those for all the target OTUs. The alignment of the representative reads in the dataset and reference sequences was performed using MAFFT (v7.505) with the settings—maxiterate 1000 -localpair (*Katoh & Standley, 2013*). The alignment results were further processed using trimAl (v1.4) with the setting -automated1-phylip (*Capella-Gutiérrez, Silla-Martínez & Gabaldón, 2009*). For construction of a Maximum-Likelihood phylogenetic tree of the 16S rRNA genes, we used IQ-TREE 2 (v2.2.0) with the settings -m MFP -B 1000 -alrt 1000 -T AUTO (*Minh et al., 2020*). The resulting phylogenetic trees were visualized using iTOL (https://itol.embl.de/).

## RESULTS

### *In situ* water sample collection

When sampling from YBH (Fig. 1B), we used sensors to measure temperature, pH, and dissolved oxygen (DO) concentration in the blue hole. The temperature, pH and DO decreased with increasing depth in the water column (Fig. 2). The temperature of water declined from about 28 °C at the surface layer to about 16 °C at the bottom layer, and the pH values dropped from 7.92 to 7.52 from the surface to the bottom layers. The surface DO of about 4.64 mg/L (71.4% saturation) was similar to that of the seawater outside the blue hole as previously reported (*He et al., 2020*; *Xie et al., 2019*). It began to decrease rapidly at about 80 m until less than 1 mg/L below 105 m (undetectable by the sensor), indicating the upper boundary of the anoxic zone in YBH. Hydrogen sulfide was close to zero between 0 and 100 m, similar to the water column out of the blue hole (*Xie et al., 2019*). With the depth increase, the concentration of $H_2S$ increased rapidly from 120 m, reaching a

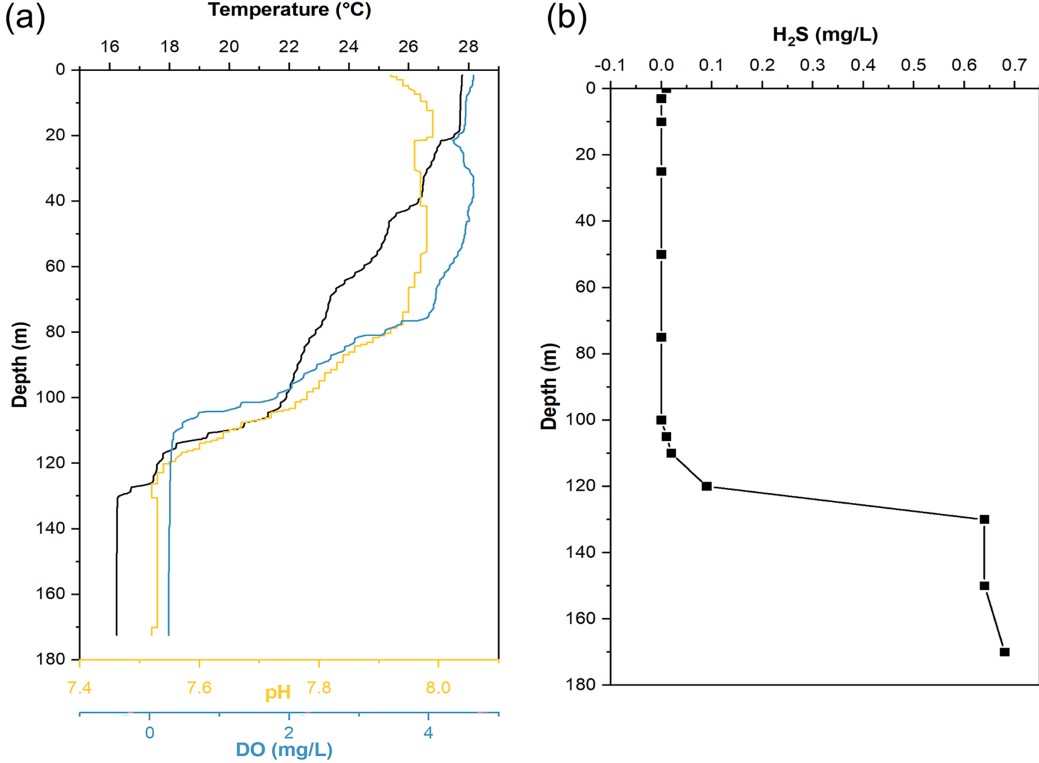

**Figure 2 Hydrological parameter profiles in the Yongle blue hole.** Profiles of seawater temperature, pH, dissolved oxygen (DO) (A), and H₂S (B) inside the Yongle blue hole were measured in April 2021.

maximum of 0.68 mg/L. Therefore, YBH has a sharp chemocline and sulfidic bottom water body. The overall stable water environment except for the depth of the oxic-anoxic interface is consistent with previously reported (*Xie et al., 2019*).

The ISMIFF successfully filtered the water of 16 layers in this cruise and obtained *in situ* filtration membranes from almost the full depth (1–290 m) of YBH (Table S1). The ISMIFF samples from different depth horizons were used to generate 17 metagenomes. In addition, one metagenome was obtained from the sample collected by a Niskin bottle at 140 m (Table S1), which was identified as an anoxic zone by the water environmental survey. A total of 82.5 Gbp metagenomics data were recovered from all the samples.

## Community composition of prokaryotes in YBH

16S rRNA gene fragments (16S miTags) were extracted from clean Illumina reads of the 17 metagenomes to decipher the microbial communities of YBH. A total of 298,488 16S miTags were obtained, and at least 3,462 16S miTags for each of the samples (Table S3). The number of miTags recovered from Illumina reads was considered sufficient to capture community composition patterns (*Caporaso et al., 2011*; *Logares et al., 2014*). After removing the bacterial and archaeal OTUs that were not classified into phyla, we identified 13 dominant prokaryotic phyla (relative abundance >5%) that were consistently detected in all the Niskin and ISMIFF samples. These phyla included Proteobacteria,

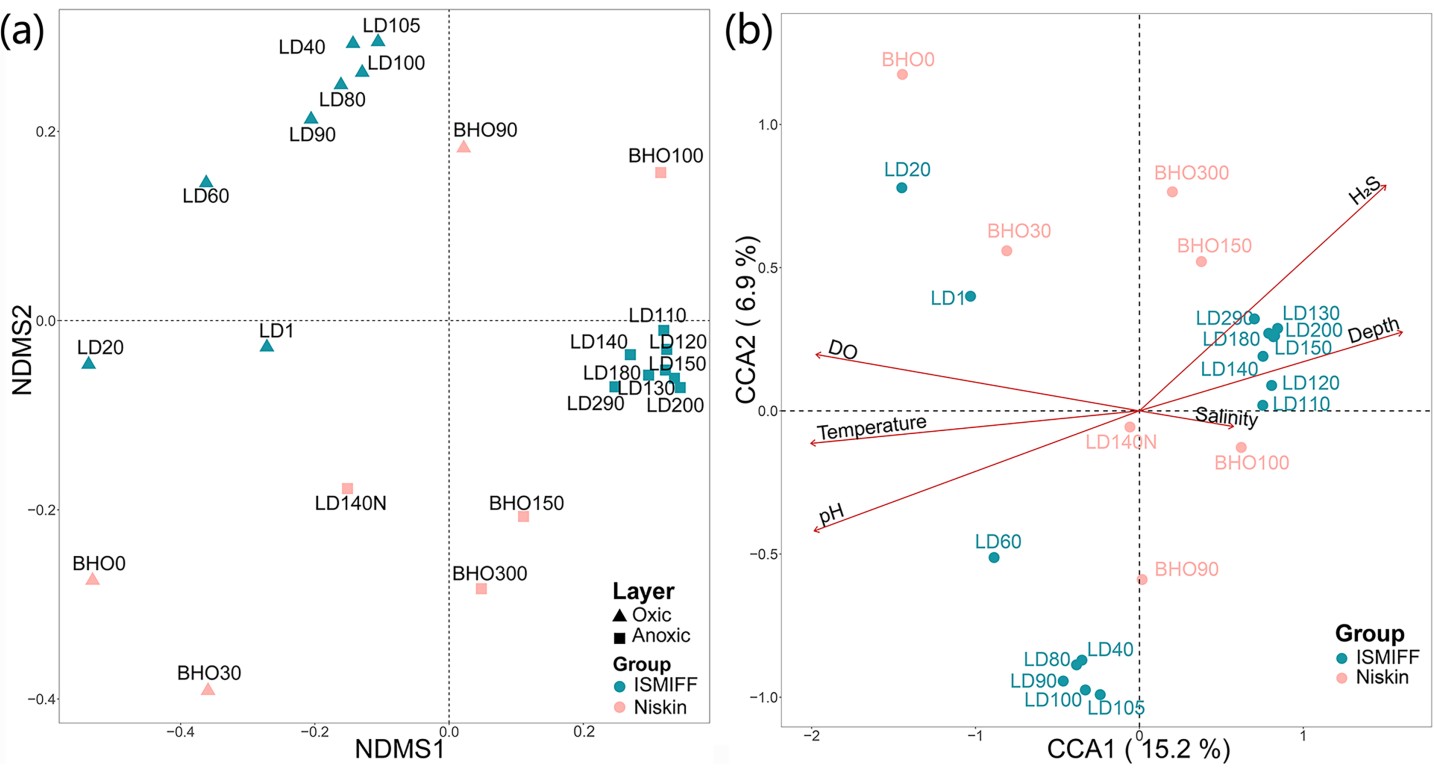

**Figure 3 Prokaryote community structure in 16 ISMIFF samples and 7 Niskin metagenomes at order level.** Water samples were collected by ISMIFF apparatus (A) and Niskin bottles (B). The orders that accounted for less than 2% of the OTUs for all samples were grouped into "minor group". Sample IDs refer to Tables S1 and S2. Sampling depth is indicated by the number in an ID. LD140N is the only Niskin sample collected in 2021, and the others were collected by *He et al. (2020)*.

Desulfobacterota, Patescibacteria, Crenarchaeota, and Bacteroidota (Fig. S1). They were further divided into 19 major orders (relative abundance >5%) (Fig. 3A). At depth of 140 m, the relative abundance of Proteobacteria in the Niskin sample (LD140N) was higher than that in the ISMIFF counterpart (LD140). This is largely due to the high abundance of Thiomicrospirales (38.13%) and Alteromonadales (26.59%) of Gammaproteobacteria in the Niskin sample (Fig. 3B). Thiomicrospirales were highly abundant in the five ISMIFF samples from the oxic area, but not those from the anoxic area. The exception is the Niskin sample in the 140 m water layer, which was much more similar to the ISMIFF samples from the oxic layers in terms of prokaryotic community structure.

Microbial communities along the water column from the sea surface to the bottom of the YBH have been previously analyzed using Niskin samples (*Peng et al., 2018*; *Yao et al., 2020*). To further compare the dominant groups between the two sampling methods with more Niskin replicates, six published metagenomes of Niskin water samples collected from YBH in May of 2017 (*He et al., 2020*) were downloaded (Table S2). Likewise, 16S miTags were extracted from the reference metagenomes and compared with the 16 ISMIFF samples (Fig. 3B and Fig. S1). At the phylum and order levels, the dominant groups of

**Figure 4 NMDS and CCA of prokaryotic communities in the samples collected by ISMIFF and Niskin bottle.** NMDS were calculated by Bray-Curtis distance for the relative abundances of the identified orders in the prokaryotic communities of YBH (stress = 0.12) (A), and the relationship between environmental factors and microbial community structures was used for the CCA ordination analysis (B). The sample IDs refer to Tables S1 and S2. LD140N is the only Niskin sample collected in 2021, and the others were collected by *He et al. (2020)*.

Niskin and ISMIFF samples were significantly different (Spearman Test; *p* = 0.0001). The most abundant order in the reference metagenomes was Alteromonadales, which also dominated the Niskin sample LD140N. In addition, the Thiomicrospirales mentioned above in the oxic layers of the ISMIFF samples did not show high relative abundance in the reference metagenomes, both in the oxic and anoxic regions. Furthermore, Campylobacterota was also one of the dominant phyla in the Niskin sample, while the other groups showed lower abundances in both the oxic and anoxic layers. The Campylobacterota was exclusively constituted by Arcobacteraceae family from Campylobacterales order (11.74%) and could not be subdivided to genus level.

NMDS plotting showed a divergence of microbial communities between the Niskin and the ISMIFF samples (Fig. S2). All the ISMIFF samples clustered together and stayed away from the Niskin samples, indicating an obvious effect of the sampling method on the relative abundances of the microbial communities. The LD140N does not cluster with the ISMIFF samples, and the microbial community structures in the samples collected by different sampling methods at the same depth were quite different. Using the relative abundances of the prokaryotic communities at order level in the 23 YBH samples (Fig. 4A), the NMDS showed that there were significant differences in the community structure of samples from different sampling methods (ANOSIM statistics, R = 0.29, *p* = 0.01).

 8/18

In addition, the samples from anoxic and oxic layers could be clearly separated but those from different cruises and methods were not proximate to each other. Then, CCA was employed to reveal the relationships between microbial community structure and environmental factors pH, DO, temperature, depth, salinity and $H_2S$ content (Fig. 4B). For the microbial communities in the hole, the two CCA dimensions explained 22.1% of the total variance, with pH, DO, temperature, and $H_2S$ having a more important effect on the community structure of these samples. In contrast, for the ISMIFF samples in the anoxic, $H_2S$ content was a more important factor affecting community structure than the others.

We plotted rarefaction curves of the observed OTUs in the 16S miTags from the V4 region (Table S3 and Fig. S3A). The increment of the number of OTUs for the ISMIFF samples was faster than that for the Niskin samples in the same batch of samples. Similarly, the number of OTUs in ISMIFF samples increases faster than in the Niskin samples at similar depths. The Shannon index was then used to estimate the diversity of the microbial communities (Fig. S3B), which showed that the ISMIFF samples were associated with a higher Shannon index than the Niskin samples (U test, $p < 0.05$). This suggests that ISMIFF could probably collect more different microorganisms in the water column for the investigation of microbial diversity and community structure (Table S1).

As previously described, Alteromonadales, Thiomicrospirales, and Campylobacterales dominate the Niskin samples, while Desulfatiglandales was most prevalent in the ISMIFF samples in the anoxic zone. To confirm their taxonomic status, we reconstructed a phylogenetic tree using the OTUs representing the most abundant of the respective orders in the 16S miTags (Fig. 5). The reference sequences for cultivated strains were selected from the NCBI database. These OTUs from the V4 region of the 16S rRNA genes were all adjacent to known strains with a high bootstrap value.

## DISCUSSION

In the present study, 16S miTags were used to reveal the prokaryotic community structure in the water column inside the YBH. The relatives of the main OTUs of Desulfatiglandales, Thiomicrospirales, Campylobacterales and Alteromonadales are sulfur-reducing bacteria (*Desulfatiglans*) (*Cifuentes et al., 2021*; *Li et al., 2021*; *Xu et al., 2022*), sulfur-oxidizing bacteria (*Thiomicrorhabdus*, *Arcobacter*) (*Magnuson et al., 2021*; *Scott et al., 2018*; *van Erk et al., 2021*), and *Alteromonas*, which confirms the metabolic potentials using their classification against SILVA database at order level (Fig. 5). We exhibited remarkably different prokaryotic community structures of the samples obtained by ISMIFF across the YBH depth, compared with our Niskin samples and those of previous studies (*He et al., 2020*). A possible interpretation of this result is that exposure of the Niskin sample to air during onboard filtration facilitated the proliferation of some $H_2S$ oxidizing and temperature-sensitive bacteria due to the drastic environmental changes (*Suter et al., 2017*; *Wang et al., 2016*). We took about 2 h to filter 40 L Niskin water. During the filtration process, the water will be exposed to air with high oxygen content. Oxygen might stimulate some aerobic bacteria to multiply rapidly (*Couvert et al., 2019*; *Kalvelage et al., 2011*), which possibly resulted in biased species richness in the microbial community inhabiting

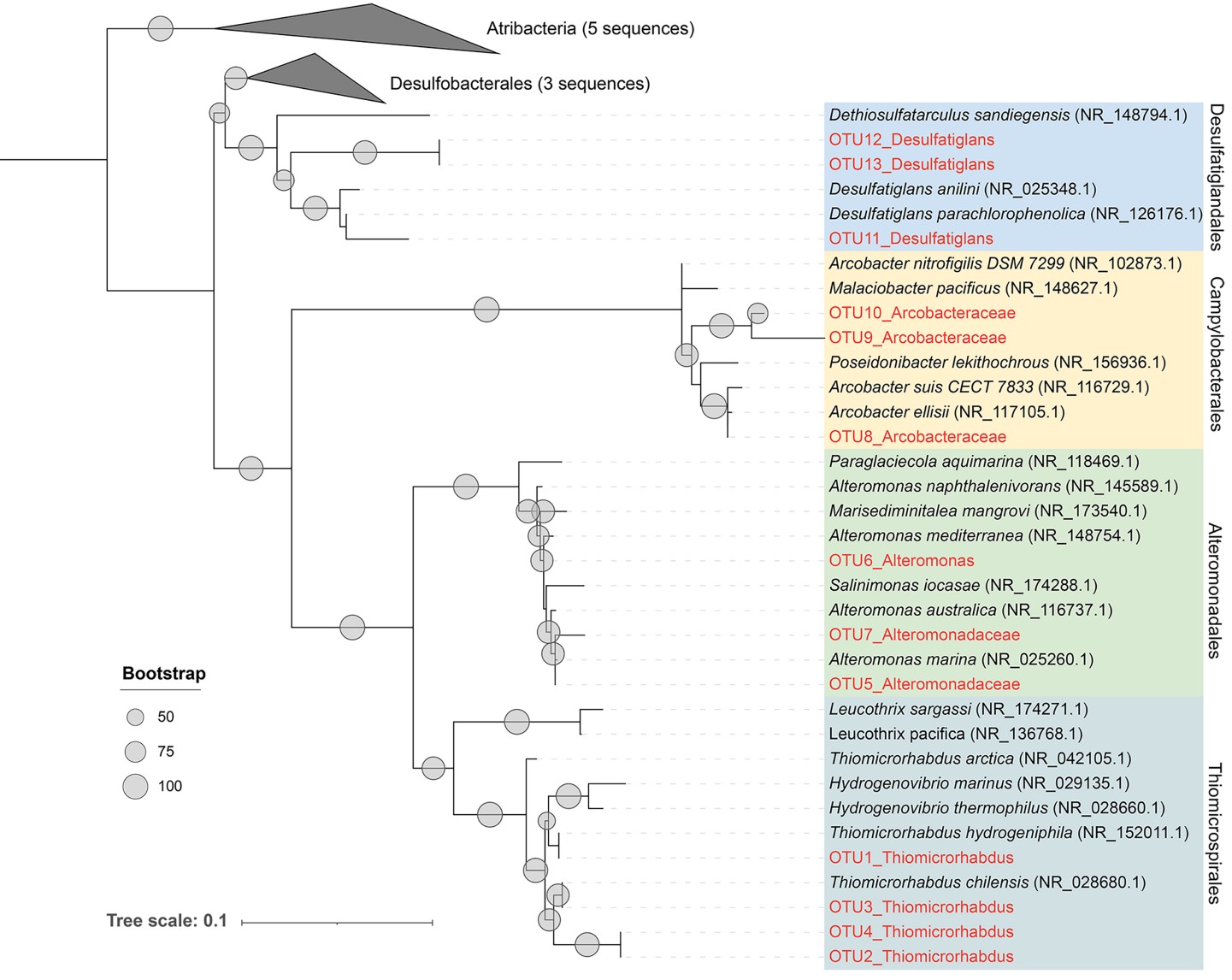

**Figure 5 Maximum-likelihood (ML) phylogenetic tree of dominant OTUs in the Niskin sample LD140N and ISMIFF sample LD140.** The representative sequences of the dominant OTUs based on 16S V4 miTags were aligned with the closest references from NCBI for the construction of ML tree. Bootstrap values above 50 based on 1,000 replicates are shown as dots with different sizes.

the YBH anoxic layer. The most abundant genus *Thiomicrorhabdus* among the Thiomicrospirales of the Niskin sample is known to be able to oxidize sulfur (*Magnuson et al., 2021*). According to the sulfide concentration, the microbe-mediated sulfur oxidation reaction mainly occurred in the oxic region. Thiomicrospirales in the oxic region in the ISMIFF samples are mainly SUP05 cluster, which can rely on oxygen to oxidize $H_2S$ that might have overflowed from YBH (*Huang, Gilna & Li, 2009*). It has been reported that denitrifying and sulfur-oxidizing *Thiomicrospira* (Thiomicrospirales) was dominant in the Niskin samples from the anoxic zone of YBH (*Wakeham, 2020*; *Zhang et al., 2021*); however, *Thiomicrospira* was much lower (on average, 0.47%) in our ISMIFF samples from the anoxic zone of YBH. The presence of the ~200 m sulfidic zone in the anoxic layer
(*He et al., 2020*) suggests that the low abundance of *Thiomicrospira* in the anoxic zone in ISMIFF samples may be more in line with the environmental trend. *Thiomicrospira* can grow efficiently under elevated $O_2$ concentrations (maximum $\mu = 0.25$ hr$^{-1}$) (*Houghton, Foustoukos & Fike, 2019*). The increase of oxygen concentration during filtration likely promotes the rapid growth of *Thiomicrospira* in the Niskin samples.

In addition, Alteromonadales in the Niskin sample also showed high abundance, and the genus *Alteromonas* (Alteromonadales) were predicted to perform denitrification in the anoxic zone (*He et al., 2020*). *Alteromonas* belongs to particle-associated microaerophilic and temperature-sensitive bacteria (*Roth Rosenberg et al., 2021*), which could rely on organic matter and nitrogen sources ($NO_3^-$) for survival (*Wei et al., 2021*). However, *Alteromonas* can probably grow rapidly when a stronger electron acceptor, oxygen, is available during water filtration onboard under a higher temperature (*Bowman & McMeekin, 2015*). In previous studies, *Arcobacter* species belonging to this family occupied the dominant position in the anoxic water column of YBH at depths of 100–150 m (*He et al., 2020*; *Zhang et al., 2021*) and they have been reported to be able to participate in sulfur cycling as a sulfur oxidizer (*Callbeck et al., 2019*; *van Erk et al., 2021*). However, sulfur oxidation mainly occurs in the oxic layer or the hypoxic layer, and the anoxic layer mainly performs sulfate reduction and denitrification (*He et al., 2019*). Other genera of Campylobacterales, such as *Sulfurimonas*, are also involved in chemoautotrophic denitrification (*Frey et al., 2014*; *He et al., 2021*; *Li et al., 2021*). In contrast, Campylobacterales showed a relatively low relative abundance in ISMIFF samples from the anoxic layers.

A large community of sulfate-reducing bacteria (SRB) has been confirmed in the bottom layer of the blue hole in this study, contributing to a sulfide-rich zone of about 170 m in YBH. The remarkable enrichment of Desulfatiglandales and Desulfobacterales can indicate that they might be present as sulfur-reducing agents in the sulfidic environment of the anoxic layer. Desulfobacterales and Desulfatiglandales are always underestimated in abundance in oxygen minimum zone (OMZ) waters (*Suter et al., 2017*), which may be the result of the settling of SRB-bearing particles in Niskin samples (*Torres-Beltran et al., 2019*) or perhaps due to physiological suppression of SRB in air. Furthermore, the prokaryotic community structures of ISMIFF samples showed a clear depth-dependent trend in the oxic zone, while in the anoxic zone, the community structures resembled to each other (Figs. 3 and 4). All the Niskin samples showed much higher variability in community structure than the ISMIFF counterparts.

In this study, ISMIFF used a 30-μm membrane to prefilter the larger particles and eukaryotes and then applied a 0.22-μm membrane for further filtration. In contrast, the reference samples for this study were directly filtered through a 0.22-μm membrane. It has also been confirmed that size fractionation has a remarkable impact on microbial community structure (*Torres-Beltran et al., 2019*). The differences between ISMIFF and Niskin samples are also related to various factors such as the depth and volume of the samples. The limitation of only one Niskin sample for metagenomic data from our cruise may affect the comparability of methods. However, previous studies have already compared the microbial community structures between deep-sea water samples collected

using Niskin and ISMIFF, which indicates that ISMIFF sampling provides a better representation of the *in situ* microbial composition (*Wang et al., 2019*). Therefore, the main purpose of this study is to emphasize the indispensability of ISMIFF for sampling in the Yongle Blue Hole. Additionally, it should be noted that the sequence data in this study only provide information about microbial community composition and cannot directly reflect the advantages and limitations of the sampling method itself. A more comprehensive evaluation would require a further consideration of other factors. Considering the stable environment in YBH, the sampling time will not be a critical factor affecting the microbial community structures. Therefore, we believe that the samples obtained in 2017 can serve as a supplement to our Niskin sample for a comparison to the ISMIFF samples. In addition, we also argue that some part of the differences in community structures might be accounted for water volume and primer selection for amplicons of the Niskin samples of the previous work.

Other blue holes and typical anoxic water columns like the Black Sea have similar sulfide-rich environments (*Cabello-Yeves et al., 2021*). In the anoxic and sulfidic environment in deep water such as the anchialine blue holes (*Gonzalez et al., 2011*; *Iwanowicz et al., 2021*), the Amberjack blue hole (*Patin et al., 2021*), and the Black Sea, sulfate reduction was mainly carried out through sulfate-reducing Deltaproteobacteria such as the genera of *Desulfobacula*, *Desulfatiglans*, and *Desulfobacter*. In the anoxic zone, the highest abundance of Amberjack blue hole is Woesearchaeota (*Patin et al., 2021*), known as active protein-utilizing archaea in the hypoxic and sulfidic environment of the Black Sea (*Cabello-Yeves et al., 2021*; *Suominen et al., 2021*). They were also abundant in the ISMIFF samples from the anoxic layers of YBH, while in low relative abundance in the Niskin samples (Fig. 3). We showed similar physiochemical profiles and microbial communities in the semi-enclosed YBH, but also remarkably distinct features such as nitrate-reducing SAR11 (*Cabello-Yeves et al., 2021*; *Ruiz-Perez et al., 2021*), compared with other oxygen-deficient water columns. This probably resulted from the sampling method and size fractions that might affect the relative abundance of dominant bacteria (*Torres-Beltran et al., 2019*). We need more efforts for establishing standardized and reproducible techniques that enable cross-scale comparisons and more accurate quantification of *in situ* activities and community structure of microbial inhabitants.

## CONCLUSIONS

In this study, we revealed the prokaryotic communities in YBH and compared with the results of different sampling methods. The community structures in the present study are more consistent among anoxic or oxic layers. Compared with traditional sampling methods, ISMIFF can filter a larger volume of water to collect more microorganisms, and therefore likely exhibit higher biodiversity in the samples as shown previously (*Gao et al., 2019*; *Wang et al., 2019*; *Wei et al., 2020*). In this study, we did not measure the environmental changes in the Niskin water and monitor the microbial community shift at the different stages of the sample processing. More lines of evidence will be required to examine the effects of sample treatments on microbial community structure (*Edgcomb et al., 2016*; *Gao et al., 2019*; *Suter et al., 2017*; *Torres-Beltran et al., 2019*). Altogether,

ISMIFF is a reliable and efficient sampling tool for microbial research of anoxic zones, which is a prerequisite for understanding the metabolic activities and adaptation strategies of YBH inhabitants and also provides unprecedented possibilities for *in situ* microbial research of anoxic water body that expands globally. Further genomics and transcriptomics studies of these OTUs will provide evidence for their metabolic activities in YBH.

## ACKNOWLEDGEMENTS

We are grateful to Sansha Trackline Institute of Coral Reef Environment Protection for their help in sample collection.

### Funding

The authors received no funding for this work.

### Competing Interests

The authors declare that they have no competing interests.

### Author Contributions

- Hongxi Zhang conceived and designed the experiments, performed the experiments, analyzed the data, prepared figures and/or tables, authored or reviewed drafts of the article, and approved the final draft.
- Taoshu Wei performed the experiments, authored or reviewed drafts of the article, and approved the final draft.
- Qingmei Li performed the experiments, authored or reviewed drafts of the article, and approved the final draft.
- Liang Fu conceived and designed the experiments, authored or reviewed drafts of the article, assisted with sample collection, and approved the final draft.
- Lisheng He conceived and designed the experiments, authored or reviewed drafts of the article, and approved the final draft.
- Yong Wang conceived and designed the experiments, authored or reviewed drafts of the article, and approved the final draft.

### Data Availability

16S miTags of YBH are available at the National Centre for Biotechnology Information (NCBI) Sequence Read Archive (SRA): PRJNA900714.

### Supplemental Information

Supplemental information for this article can be found online at http://dx.doi.org/10.7717/peerj.16257#supplemental-information.

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
