# Peer review of "Metagenomic 16S rDNA reads of in situ preserved samples revealed microbial communities in the Yongle blue hole"

_PeerJ, doi:10.7717/peerj.16257_

## Round 0.1 · original submission · Major Revisions

PeerJ recognizes the value inherent in both the unique environment studied and the use of novel sampling equipment and comparisons between methods in understanding the system. However, both reviewers raised significant issues which should be addressed in a revised version point by point and thoroughly. I agree with the vast majority of the reviewers' comments, and particularly agree with the perspective raised by R2 that the scope of the sampling design lends itself better to a rewrite of the manuscript to focus more on understanding the microbial ecology of the system and less on the methodological comparisons; where the methodology comparison informs questions or suggestions for further study this is valuable. As such please revise according to R2's "second option". We look forward to receiving an extensive response to reviewers along with your revised manuscript.

·

Basic reporting

The study is well-written and references appropriate studies. There is appropriate background and relevant studies are referenced. However, I see no mention of sequence data availability, either in NCBI SRA or another public database.

Experimental design

The study is well designed and the appropriate analyses were applied. There is information missing from the Methods section necessary to replicate the study (see full comments below).

Validity of the findings

The findings are valid, but the experiment is not sufficient to draw technical conclusions about the ISMIFF sampling apparatus because only one Niskin sample could be directly compared with 16 ISMIFF samples.

Additional comments

The authors present a study on the microbial community composition of the Yongle Blue Hole (YBH) water column. It follows up on previous studies, which used amplicon sequencing and targeted PCR of functional genes to draw conclusions about the microbial ecology and biogeochemistry. Here, the authors performed shotgun metagenomic sequencing and extracted marker genes (“miTags”) from the metagenomes to determine community composition. The samples were collected with an autonomous instrument, the ISMIFF, which filtered 28-68 L water (except for one outlier sample of 5 L) and preserved the filters in situ. One sample was collected in a “traditional” manner with a Niskin bottle and shipboard filtration. This sample, along with sequence data from previous YBH studies using traditionally collected samples, was compared with the ISMIFF samples to characterize the YBH microbial community and make inferences about the benefits of autonomous sampling.

Blue holes are fascinating environments with unique microbial communities. We can learn a lot about redox chemistry and microbial biology in low-oxygen, sulfidic environments from studying habitats like the Yongle Blue Hole. Moreover, the ISMIFF autonomous sampler is a promising instrument that, like other autonomous sampling instruments, shows great potential for sampling marine water columns. The ability to filter up to 70 L water in situ is exciting and will certainly improve our understanding of microbiomes in blue holes.

The authors did an admirable job extracting ribosomal RNA marker genes from the metagenomes and summarizing their findings from the YBH. However, considering their previous studies from this environment, I don’t understand why the metagenomes were not leveraged to their fullest extent to draw conclusions not only about taxonomy but also microbial function. A few examples of other potential work include: assessing taxonomic composition using all reads (e.g., with a tool like sourmash or Kraken2), assembling reads and using the assemblies to predict and annotate open reading frames, and binning genomes (using aforementioned assemblies). Assessing community composition using marker genes could just as easily have been done using amplicon sequencing, and indeed, has already been done in previous studies on the YBH.

Moreover, the comparative aspect of the study is not robust. A comparison of one Niskin sample with 16 ISMIFF samples can’t be used to draw conclusions about differences between autonomous and traditional sampling. The use of previously published sequence data, while useful, can’t be used as a direct comparison with ISMIFF because they were collected more than four years apart. While many of the proposed advantages to in situ filtration presented here may be valid, without sufficient replication they are just speculation and are better left to a study that specifically tests these hypotheses.

Despite these drawbacks, I believe the information presented in this study is important and I recommend publication with minor revisions. These include:

Methods: There is no information on how the phylogenetic tree in Figure 5 was generated. How were the “dominant” taxa chosen? What alignment method was used? What software generated the tree?

Lines 263-272: The Figure 5 caption says the sequences in the tree are only from the Niskin sample, but this paragraph doesn’t state that. Please clarify.

Figure 3: The bars representing 2017 samples need to be clearly labeled as such so that readers know they were collected in a different study.

Figure 4: Same comment for the 2017 samples vs the single 2021 Niskin sample. Since the ordination points are already classified by shape and color, a clear description in the figure legend should identify “LD140N” as the only 2021 Niskin sample.

Figure 5: This is a nice tree, but why is there only one for the Niskin sample? It would be nice to see a similar tree for the corresponding 140 m ISMIFF sample.

Discussion: I would like to see Figures 3 and 4 described in more detail. For example, all Niskin samples show much more variability within the oxic and anoxic groupings than the ISMIFF samples. In addition, there needs to be a stronger statement on the limitations of comparing a single sample (Niskin) with 16 others (ISMIFF) and that this study does not present conclusions on the technical advantages and limitations of ISMIFF based on the sequence data.

·

Basic reporting

The language needs some attention in some parts (see detailed comments)

Experimental design

The experimental design is straight forward.

Validity of the findings

See below:

Additional comments

In the study ‘Metagenomic 16S rDNA reads of in situ preserved samples revealed microbial communities in the Yongle blue hole ‘. the authors collected 16 metagenomic samples at different water depths using an in situ microbial filtration and fixation (ISMIFF) apparatus and one additional sample collectd with a niskin bottle to provide an overview of the microbial community composition in the Yongle blue hole (YBH), one of the (or the?) deepest sinkhole we know of and therefore a very unique environment. Additionally they collected conductivity, temperature, dissolved oxygen, pH, salinity, and H2S data along the depth profile. For additional reference the authors compared their findings to six previously published metagenomes of water samples collected from YBH with Niskin bottles in 2017 by He et al.

Their main conclusion is that using in situ filtration (ISMIFF) is the much better approach to identify the communities at different depths of the Yongle blue hole than using traditional NISKIN bottles. The authors attribute this to the introduction of aerobic species during the onboard filtration, required for NISKIN samples.

General comments:
Over all I think this study should be published but I do have some reservations towards the current version. My main problem is that the authors use the samples collected from a very unique location to basically conduct a method comparison. And the method comparison has some flaws that need to be better addressed.
So first of all my recommendation for this manuscript is major revisions, but to a point where most of the manuscript needs to be redone. This is not based on the fact that it is not a good study, but I think the authors need to figure out what they want to highlight here and then fundamentally restructure the manuscript.

I see this going 2 ways. For one the authors keep the focus on the differences in methodology and drive home the point that traditional Niskin bottle watersampling only provides a very limited insight; but then they will in my opinion need to do some additional sampling. The main problems are:
- only one comparable Niskin sample was taken, and the other Niskin samples were from a different study at a different time
- If I am correct than significantly different volumes of water were collected and filtered by the ISMIFF and by the Niskin approach
- And I think most problematic the one set was prefiltered and the other was not
I don’t want to say that the difference is not striking, and the authors are probably correct in saying that the Niskin approach might not reflect the real community very well, but if this is to be a method focused manuscript then these things need to be addressed.

The other option, and I would recommend that one, would in my opinion be to shift the main focus of the manuscript on the characterization of the microbial community in this very unique ecosystem. The authors could go deeper into the ecology, discuss the different species and their relation to the governing physical parameters at the different water depths, maybe even go exploring the metabolic functions. The findings of the differences in the sampling methods can still be a part and the authors should voice their concerns about the Niskin sampling method, but it should not be the dominating main focus of this study.

Additional comments:
- At some points some sentences are a bit confusing or incorrect prepositions are used.
- I don’t fully understand why only a tree from the NISKIN bottle sample and not the ISMIFF samples was created as the main point was that that sample is most likely heavily biased due to the onboard filtering.
- I don’t fully understand how the authors came to the conclusion that H2S is the most important factor.
- At some points thee language needs some attention
- 4.64 mg/L Oxygen in the overlying watercolumn seems low to me. Could you maybe include the 100% saturation? There might have been an error converting from %saturation, which is the unit measured by the optode, to the concentration (mg/L).

Minor comments:

Abstract
Line 26: The authors mention ‘reported samples’, I assume this refers to the NISKIN samples from the earlier study, but this is not clear the way it is stated right now.

Results
Line 186: The environmental parameters obtained in this study showed that the readings of these sensors decreased with increasing depth across the water column. Maybe rephrase to ‘The temperature, pH and DO decreased with increasing depth in the water column’

Line 213: Please replace ‘with the removal’ with ‘after the removal’, and maybe rephrase: (e.g., After removing the bacterial and archaeal OTUs that were not classified into phyla, we identified 22 dominant prokaryotic phyla (relative abundance >1%) that were consistently detected in all the xxx samples. These phyla included Proteobacteria, Desulfobacterota, Patescibacteria, Crenarchaeota, and Bacteroidota (Fig. S1).

Line 232: Please replace ‘no matter in’ with ‘both in the’

Discussion

Line 309: ‘Arcobacter is also anoxic denitrifiers’ is coming out of nowhere and seems to be an incomplete sentence.

Line 319: “may be the result of” instead of ‘which may be accounted for’?

Line 341: remove the word ‘here’
Figure 1:
Please, replace ‘are referred to’ with ‘are explained in’ or ‘refer to’
Figure 3:
Please, remove the word ‘in’ from ‘in of the OTUs’

---

## Round 0.2 · Minor Revisions

Both reviewers have re-reviewed this manuscript and concur that it is acceptable for publication. However, one key issue remains aloft. This manuscript alludes to upcoming or otherwise unpublished data (genomes and functional annotations) and this is inappropriate. All discussion and alllusion to these data should be removed from the manuscript, and discussion of metabolic functions facilitated by these allusions should be limited to conjecture and only in the discussion section, as these cannot be supported by the data provided. We believe this is a relatively simple revision (minor) and look forward to your final draft.

·

Basic reporting

The manuscript is clearly written with appropriate references. The raw data is available on NCBI as described.

Experimental design

The authors addressed my original concerns and he research question is now more appropriately described. Technical methods and statistical analyses are rigorous and replicable.

Validity of the findings

no comment

Additional comments

The authors addressed my previous concerns and I consider the manuscript ready for publication.

·

Basic reporting

The language and story line has significantly improved.

Experimental design

The experimental design is straight forward and valid.

Validity of the findings

See below.

Additional comments

The manuscript shows a considerable improvement. The attention to the language and the exclusion of the more technical part of the story has made the storyline more straight forward and the paper easier to read. The analysis shown is valid and consistent, though a bit on the minimalistic side, especially given the fact that full genomes are available but only the 16S RNA sequences are used. The authors state this is for the sake of comparing this study with earlier studies, but this comparison would not dictate to focuses solely on the composition. The authors state that the functional analyses were performed but for the sake of brevity these findings will be discussed in a different paper. I guess this is valid, but still reduces this manuscript to a very minimal contribution to understanding these unique ecosystems with not much insight in ecological or metabolic functions to be gained. I feel that it still would maybe have been better to not split the functional analysis and the taxonomy data in two manuscripts but combine them into one coherent story.
In the end, given the unique environment and the new approach that revealed a clear difference to earlier studies, this study will contribute to understanding microbial community dynamics in these specific and unique ecosystems, and should therefore be published

---

## Round 0.3 · accepted · Accept

Thank you for your revisions and we feel that the manuscript is now ready for publication in PeerJ.